# Complexing the Marine Sesquiterpene Euplotin C by Means of Cyclodextrin-Based Nanosponges: A Preliminary Investigation

**DOI:** 10.3390/md20110682

**Published:** 2022-10-29

**Authors:** Alessandra Bertoli, Anthea LoBue, Luca Quattrini, Stefania Sartini, Beatrice Polini, Sara Carpi, Francesco Paolo Frontini, Graziano Di Giuseppe, Graziano Guella, Paola Nieri, Concettina La Motta

**Affiliations:** 1Department of Pharmacy, University of Pisa, Via Bonanno 6, 56126 Pisa, Italy; 2Marine Pharma Centre, University of Pisa, Via Bonanno 6, 56126 Pisa, Italy; 3Myocardial Infarction Research Laboratory, Department of Cardiology, Pulmonology, and Angiology, Medical Faculty, Heinrich-Heine-University, Universitätstraße 1, 40225 Düsseldorf, Germany; 4Department of Surgical, Medical and Molecular Pathology and Critical Care Medicine, Via Paradisa 2, 56124 Pisa, Italy; 5NEST, Istituto Nanoscienze-CNR and Scuola Normale Superiore, Piazza San Silvestro, 56127 Pisa, Italy; 6Department of Biology, University of Pisa, Via Luca Ghini 13, 56126 Pisa, Italy; 7Laboratory of Bioorganic Chemistry, Department of Physic, University of Trento, Via Sommarive 14, 38050 Povo Trento, Italy

**Keywords:** euplotin C, β-cyclodextrin, β-cyclodextrin-based nanosponges, SPE, ATR-FTIR analysis, NMR analysis, ESI-MS analysis, LC-DAD-ESI-MS analysis

## Abstract

Euplotin C is a sesquiterpene of marine origin endowed with significant anti-microbial and anti-tumor properties. Despite the promising functional profile, its progress as a novel drug candidate has failed so far, due to its scarce solubility and poor stability in aqueous media, such as biological fluids. Therefore, overcoming these limits is an intriguing challenge for the scientific community. In this work, we synthesized β-cyclodextrin-based nanosponges and investigated their use as colloidal carriers for stably complex euplotin C. Results obtained proved the ability of the carrier to include the natural compound, showing remarkable values of both loading efficiency and capacity. Moreover, it also allowed us to preserve the chemical structure of the loaded compound, which was recovered unaltered once extracted from the complex. Therefore, the use of β-cyclodextrin-based nanosponges represents a viable option to vehiculate euplotin C, thus opening up its possible use as pharmacologically active compound.

## 1. Introduction

Nature is an endless source of functionally active compounds, which can be profitably exploited by medicinal chemists for the development of novel and effective drugs. This is the case of euplotin C (EC, Figure 1), a sesquiterpenoid derivative endowed with a significant cytotoxic activity, which is produced by the marine ciliated protist *Euplotes crassus* to participate in the conquest and maintenance of its ecological niche over time [1]. Since the early 1990s, EC has become the object of several investigations carried out to verify its possible use as drug. Thanks to these studies, the compound proved to be effective against pathogenic protozoans belonging to the *Leishmania* genus [2,3,4]. It also exhibited an anti-tumor profile, showing significant cytotoxic and pro-apoptotic activities when tested in vitro against several tumor-derived cell lines [5,6]. In particular, EC displayed relevant activity towards the human cutaneous melanoma A375, 501Mel and MeWo cell lines, with a potency about 30-fold higher than that observed in non-cancer cells. Therefore, it might be deemed a promising anti-melanoma agent [7].

However, despite its efficacy, EC never became a full-fledged drug. All the attempts to advance the compound to pre-clinical and even clinical levels clashed with its chemical shortcomings, mainly represented by both scarce solubility and poor stability in aqueous media, such as biological fluids [8,9]. Indeed, due to its amphiphilic character, EC gives rise to micellar aggregates unless diluted below its critical concentration. At the same time, when dissolved in the monomeric form, it shows a half-life of a few hours, undergoing a progressive chemical degradation [10].

Therefore, overcoming these limits is necessary to avoid losing an active derivative potentially useful for the treatment of still-untreatable diseases. To this end, the use of carriers able to encapsulate and vehiculate an intact EC to the target sites comes in handy. An early attempt has been recently pursued by Guella and co-workers using heptakis(2,6-di-*O*-methyl)-β-cyclodextrin (DIMEB) [10]. EC proved to enter the hydrophobic cavity of the cyclic saccharide protruding its prenyl chain toward the narrower rim of the toroid structure. However, the acetyl residue remained outside the larger rim of the cone. Although this improved the water solubility of EC thanks to the surrounding hydrophilic crown, DIMEB did not prevent the chemical degradation of the molecule. In fact, as the latter is driven by the initial cleavage of the acetyl group [10], failing to wrap this residue inside the backbone of the carrier inevitably triggers the chain reaction leading to EC decomposition.

Taking advantage of our expertise in the field of β-cyclodextrin (β-CD) based nanosponges (NS) as valuable drug delivery systems [11], we investigated the use of this nano-structured colloidal vector to better encapsulate the target molecule preserving over time its chemical stability.

NS are synthesized by suitably cross-linking single units of β-CD, which is a cyclic, hydrophilic, non-toxic oligosaccharide consisting of seven glucose residues. The resulting hyper-cross-linked polymer is fully biocompatible, non-cytotoxic and easily obtainable in high amounts through sustainable and cost-effective synthesis. Being broadly usable at the industrial level, it is increasingly becoming the reference tool for drug delivery [12,13,14]. NSs are able to efficiently incorporate structurally different drugs within their own structure, either as inclusion- or non-inclusion-complexes. In doing so, they allow us to increase the solubility of the hosted compounds while also preserving their chemical stability. Therefore, they seem to be the ideal carrier for challenging compounds such as EC, whose therapeutic potential cannot be exploited due to low solubility and poor stability in aqueous media [15,16,17].

Moreover, unlike the single DIMEB unit, the tridimensional structure of the sponge may host the whole structure of EC which, as a result, turns out to be fully protected from the degrading activity of the medium.

In this work we describe the obtainment of a novel EC-NS complex (EC-NS), whose structural characteristics were thoroughly investigated by chromatographic, spectroscopic and spectrometric data.

## 2. Results and Discussion

The sample of EC used in the study was recovered from an *E. crassus* SSt22 strain culture by solid-phase extraction (SPE) fractionation and successive semi-preparative high performance liquid chromatography (HPLC) analysis of the SPE fractions, in order to maximize EC recovery (7.1 mg, yield 16%) from the native organic phase (45.2 mg). Once retrieved, the purified EC (99%, HPLC purity) was exploited for the obtainment of the EC-NS, accomplished by combining the natural compound with the nano-carrier in a 1:1 *w/w* ratio. Then, the obtained complex was thoroughly investigated as described hereafter.

### 2.1. Attenuated Total Reflectance-Fourier Transform Infrared (ATR-FTIR) Studies

The ATR-FTIR spectrum of the recovered EC showed a signal at 1734 cm^−1^, attributable to the stretching vibration of the carbonyl residue of the acetyl group, typical of the ester function, and an intense signal at 1564 cm^−1^, due to the ethereal vinyl portion. Signals at 2800–2900 cm^−1^ were assigned to stretching vibrations of aliphatic C–H bounds of both the tricyclic ring and the side chain, while the wide band at 3300 cm^−1^ may be attributable to a possible presence of water in the sample (Appendix A, Appendix A). The ATR-FTIR spectrum of the synthesized NS displayed a key signal at 3300 cm^−1^, attributable to the stretching vibration of the alcoholic O–H bounds, and a signal at 2900 cm^−1^, due to the stretching vibration of the aliphatic C–H bounds. An additional signal at 1748 cm^−1^ can also be seen, assigned to the stretching vibration of the carbonyl bound, while the signal at 1023 cm^−1^ is due to the stretching vibration of the alcoholic C-O bound (Appendix A, Appendix A).

Both the spectra of EC and NS were compared to those of the loaded complex EC-NS (Appendix A, Appendix A) and the 1:1 physical mixture of the two components, EC and NS (Appendix A, Appendix A). Then, a superimposition of the last two spectra was made, in order to confirm the formation of the inclusion complex (Figure 2).

In the binary mixture, both the signal at 1734 cm^−1^ (due to the acetyl group of EC) and the signal at 1023 cm^−1^ (owing to the alcoholic functions of NS) are characteristic of their respective components. In addition, an intense signal at 1635 cm^−1^ is also evident, attributable to the C=C stretching vibration of the EC side chain. On the whole, the ATR-FTIR spectrum of the physical mixture turned out to be a clear superimposition of the spectra registered for the single components, EC and NS. Indeed, neither shifts nor modifications of the absorption bands were observed, thus demonstrating the absence of significant interactions between NS and the natural compound. On the contrary, the ATR-FTIR spectrum of the EC-NS complex displayed key differences. In fact, signals related to EC are less intense. Moreover, and significantly, the signal due to the stretching vibration of its carbonyl group is shifted to 1748 cm^−1^, thus resulting in a higher wavenumber as compared to that in the spectrum of the pure EC (1734 cm^−1^). Signals characteristic of NS (1023, 2900 and 3300 cm^−1^) are more intense and show significant ATR-FTIR wavenumber shifts as compared to the corresponding ones in the binary mixture, thus confirming the inclusion of EC into the nano-carrier.

### 2.2. ESI-MS Studies

Direct-infusion ESI-MS analyses were carried out to characterize both the original NS carrier and the EC recovered from the culture. Moreover, the same technique was also used to analyze EC obtained by extraction from the loaded EC-NS complex.

All spectra were acquired in positive-ion mode.

EC sample preparation for ESI-MS analysis was carried out using MeOH. Results obtained were compared to those collected with a reference sample of pure EC. Full scan spectra were registered in the range from 100 to 500 *m*/*z* tuning EC monitoring parameters at [M + H + Na]^+^ 315.1 *m*/*z* and providing the diagnostic daughter ion at the nominal mass 254.9 *m*/*z* by MS/MS experiments (Appendix A, Appendix A). The dimeric ion [2EC + Na^+^] at 606.9 *m*/*z* and the diagnostic sodium adduct peak at 315.1 *m*/*z* were present, as reported by Guella and co-workers [8]. The protonated-molecular ion [M + H]^+^ at 293 *m*/*z* was observed only after performing the MS-MS experiment on the 315.1 *m*/*z* peak directly. Moreover, the fragment ions at 254.8 *m*/*z* ([M + Na − CH_3_COOH]^+^, 60%) and 233.3 *m*/*z* ([M + H − CH_3_COOH]^+^, 25%) were also obtained as diagnostics for the EC structure (Figure 3; Appendix A, Appendix A).

The ESI-MS analysis carried out on the NS carrier highlighted diagnostic peaks at 1157.5 *m*/*z* (100%), 1347.8 *m*/*z* and 663.5 *m*/*z* (Appendix A, Appendix A). All the ions were subjected to MS-MS experiments in order to find their characteristic fragment pathway. In particular, the ion at 1157.5 *m*/*z* afforded the following daughter ions: 995.3 *m*/*z* (100%), 833.3 *m*/*z* (70%), 671.2 *m*/*z* (49%) and 509.2 *m*/*z* (10%), all derived from consecutive losses of the neutral dehydrated glucose molecule (M − 162) (Figure 4; Appendix A, Appendix A). Indeed, fragmentation can occur casually in every acetal junction of the open β-CD, as previously reported by Sforza and co-workers [18].

The ion at 1347.8 *m*/*z* was fragmented affording the following daughter ions: 463.2 *m*/*z* (100%), 551.1 *m*/*z* (90%), 607.1 *m*/*z* (70%) and 495.2 *m*/*z* (30%), all resulting from consecutive losses of neutral molecules (56 Da) (Figure 5; Appendix A, Appendix A).

A monomeric β-cyclodextrin (β-CD) sample was dissolved in MeOH and analyzed under the same experimental conditions in order to compare its typical ions with the peaks observed in the ESI-MS study of NS. The analysis highlighted ions at 1157.5 *m*/*z* and 663.4 *m*/*z* as the most intense peaks, which it had in common with the NS sample. The first of these corresponds to an adduct between β-CD and a sodium ion [M + Na]^+^, thus demonstrating that NS was fragmented by losing whole β-CD units.

As for EC extracted from the loaded complex, two different samples were obtained by treating EC-NS with either apolar solvents (hexan:ethyl acetate (9:1), EC-NS1 sample) or polar solvents (MeOH, EC-NS2 sample).

The two samples were analyzed using direct infusion, in positive-ion mode, and MS spectra for EC-NS1 and EC-NS2 were compared to those obtained from the pure samples of EC and NS, respectively. The diagnostic ions were detected at 315.1 *m*/*z* (EC-NS1 (65%) and EC-NS2 (100%)), 606.9 *m*/*z* (EC-NS1 (98%) and EC-NS2 (50%)), 1347 *m*/*z* (only in EC-NS1 (40%)) and finally at 1157 *m*/*z* (only in EC-NS2 (80%)).

Fragmentation of the ion at 607 *m*/*z* was not possible, as it turned out to be very stable, while fragmentation of the diagnostic ion at 315.1 *m*/*z* provided the daughter ion at 254.8 *m*/*z* (100%) (Appendix A, Appendix A). A comparison between the ESI-MS spectrum of EC and the ESI-MS spectra of both EC-NS1 and EC-NS2 demonstrated that NSs are able to include the target derivative and that it is recovered unaltered once extracted from the complex with both lipophilic and polar solvents.

### 2.3. LC-PDA-ESI-MS Studies

Data produced by the direct infusion ESI-MS analyses were exploited to determine the results of an LC-PDA-MS investigation, which was carried out on both the EC-NS1 and EC-NS2 samples. Analyses of the two samples showed two chromatographic peaks with retention times of 11.1 ± 0.1 min and 4.18 ± 0.2, corresponding to the pure EC and NS, respectively (Figure 6). The observed ions at 315.1 *m*/*z* and 254.8 *m*/*z* were assigned to the loaded substance EC, while the ions at 1157.5 *m*/*z* and 663.4 *m*/*z* were assigned to the carrier NS. Once combined, the four ions were considered for the qualitative analysis of the complex EC-NS.

### 2.4. Loading Efficiency and Loading Capacity of NS

The amount of EC loaded into the carrier was determined by a LC-SRM-MS investigation on both the EC-NS1 and EC-NS2 extracts. The analyses were carried out using the same tuning parameters used for the pure EC reference substance (positive-ion mode, diagnostic ions at 315.1 *m*/*z* and 254.8 *m*/*z*, Appendix A, Appendix A).

The amount of EC in the EC-NS complex was determined by means of a suitably working curve. The result obtained allowed us to calculate a 90% loading efficiency and a 86% loading capacity of the carrier, thus confirming the relevant effectiveness of NS at interacting with the target EC.

## 3. Materials and Methods

β-Cyclodextrin (β-CD) and diphenylcarbonate, used to synthesize NS, were obtained from Sigma–Aldrich (St. Louis, MO, USA). MilliQ grade water, acetonitrile and methanol (Carlo Erba, Milan, Italy) were used for LC-PDA-MS analysis. SPE cartridges (C18, 10 g/50 mL, Phenomenex, Bologna, Italy), PTFE membrane Luer-lock filter (0.45 μm, 15 mm, Phenomenex, Bologna, Italy), TLC RP−18 glass and silica gel aluminium folium plates (F254, 5 × 10 cm, Merk, Rome, Italy) were used. ^1^H NMR spectra were recorded either in CDCl_3_-*d*_1_ (EC) or in D_2_O solution (NS and EC-NS), on a Bruker 400 spectrometer operating at 400 MHz (Bruker, Wien, Austria). ATR-FTIR spectra were obtained with an Agilent Cary 620 FT-IR Microscope Spectrometer (Santa Clara, CA, USA). The LC system consisted of a Thermo Finnigan Surveyor liquid chromatograph pump equipped with a Thermo Finnigan Photodiode Array Detector and an ion trap LCQ Advantage mass spectrometer (Thermo Finnigan LLC, San Jose, CA, USA).

### 3.1. Chemical Synthesis of β-CD-NS

β-CD-NSs were synthesized following a previously reported procedure [11]. Briefly, a mixture of β-CD (1.00 mmol) and diphenylcarbonate (4.00 mmol) was heated under stirring at 90 °C for 5 h. Once the reaction was completed, the resulting cross-linked solid was ground in a mortar and washed with water and ethanol. The white product (NS) so obtained was anhydrified in a vacuum oven, then characterized through spectral data (Appendix A, Appendix A) [11].

### 3.2. Euplotin C Isolation from Cultures of Ciliated Strains

The target EC was isolated from cultures of *E. crassus* according to the sequential steps summarized in Appendix A (Appendix A). The SSt22 strain of the ciliate protist was cultured as previously reported [2]. After discarding the culture fluid, a collection of cell pellet samples (11 cell pellet lots, total cell volume 2.7 mL, ca. 1.08 × 10^8^ cells) was centrifugated and re-suspended in absolute EtOH. After agitation and sedimentation, the ethanolic supernatant was collected. Then, each cell pellet was recovered and further extracted with EtOH (5 mL × 3 times), and the resulting extracts were gathered together and added to the ethanolic supernatant. After concentration by rotavapor at 30 °C, the ethanolic phase was partitioned between n-hexane-AcOEt (8:2) and water (10 mL × 5 times). The organic phase was recovered, then evaporated under vacuum at 30 °C to provide an organic residue (45.2 mg) which was fractionated as described hereafter.

### 3.3. Euplotin C Purification from Organic Extracts

The organic residue resulting from cell-pellet extraction was fractionated by SPE C−18E (Strata Phenomenex, 500 mg/6 mL, 55 μm, 70Å) eluting by MeOH-Water (80:20) and MeOH. Six different fractions were collected (A–F), which were analyzed and compared to a pure reference sample of EC by using an RP-C18 TLC glass plate (Merck Silica gel F254, 5 × 10 cm), MeCN-Water (7:3) as the eluent, and sulfuric vanillin 1% and cerium sulfate 10% as the spray reagents. TLC screening showed fractions A (6.0 mg), B (5.3 mg) and C (2.7 mg) as mixtures containing the target EC (R_f_ 0.197). Fractions D, E and F did not show any spot related to the reference EC; therefore, they were discarded (Appendix A, Appendix A). A further SPE fractionation step was then carried out on the A–C fractions in order to increase the recovery of pure EC. The resulting samples were gathered and finally purified by semi-preparative HPLC analysis, using a Waters 600E pump, Waters UV 400 detector (manual injection Rheodyne valve, 100 μL loop) equipment, a Phenomenex Fusion RP−18 column (10 × 150 mm, 4 μm,) and isocratic chromatographic conditions (CH_3_CN-Water, 7:3; 1.2 mL/min flow rate). The chromatogram was registered during the whole run (40 min), at 215 nm, providing 7.1 mg of pure EC (Appendix A, Appendix A). An aliquot of the purified EC sample (99%, HPLC purity) was finally characterized by ATR-FTIR (Appendix A, Appendix A), ^1^H-NMR (Appendix A, Appendix A), ^13^C-NMR (Appendix A, Appendix A) and ESI-MS multi-profiling data (Appendix A) as previously described, in order to confirm the structural features of the target sesquiterpene.

### 3.4. Preparation of Binary Mixture of EC and NS

A binary physical mixture of EC and NS was prepared by mixing the two solid components in a glass mortar, in a 1:1 weight ratio.

### 3.5. Preparation of EC-loaded NS (EC-NS)

The loaded NSs were prepared at a NS:compound ratio of 1:1, *w*/*w*. Appropriate amounts of the two components were added with anhydrous EtOH and left under stirring at room temperature for 1.0 h. Then, the solvent was evaporated under reduced pressure and the resulting solid was washed with hexan to remove the unloaded EC, then anhydrified in a vacuum oven.

### 3.6. Recovery of EC from the Loaded NS

The loaded complex EC-NS (1.5 mg) was first extracted with a mixture of *n*-hexane/ethyl acetate (9:1 ratio, 5 × 500 μL), and then with MeOH (5 × 500 μL). During each extraction step, the EC-NS suspension was shaken by vortex for 1 min and left to macerate for 24 h at room temperature and in dark light conditions. The supernatants were collected and evaporated by rotary evaporator at 30 °C, providing in turn an apolar extract (EC-NS1, colourless, 1.0 mg) and a polar extract (EC-NS2, colourless, 0.3 mg). Both EC-NS extracts were stored in a freezer (−20 °C) prior to sample preparation for the subsequent ESI-MS and LC-PDA-ESI-MS analyses, as previously described.

### 3.7. ESI-MS Analysis

The obtained pure EC (1.1 mg), monomeric β-CD (1.2 mg), NS (1.5 mg) and EC extracted from the loaded complex EC-NS (1.1 mg) were diluted in MeOH, filtered by Luer-lock syringe filter (PTFE membrane, 0.45 μm, 15 mm, Phenomenex) and analyzed by ESI-MS analysis (syringe flow rate 10 μL/min, 500 μL, sheath gas flow-rate 62 arbitrary units, auxiliary gas flow 9 arbitrary units, capillary voltage −16 V, positive mode and capillary temperature 280 °C). The obtained diagnostic and daughter ions are summarized in Appendix A (Appendix A).

### 3.8. LC-PDA-ESI-MS Analyses

The LC-PDA-ESI-MS screening was carried out on both the pure EC and the EC-NS extract samples. The LC system consisted of a Thermo Finnigan Surveyor liquid chromatograph pump equipped with an analytical Fusion RP−18 column (150 × 4.6 mm i.d., 4 μm, Phenomenex), a Thermo Finnigan Photodiode Array Detector (detection range 200–500 nm) and an ion trap LCQ Advantage mass spectrometer. The LC-PDA-MS screening was carried out by an isocratic eluent program using CH_3_CN (solvent A) and water (solvent B) 70:30 *v/v* (50 min) (flow 0.5 μL/min, injection volume 20 μΛ) as the eluting mixture. The chromatogram peaks were registered at 215 nm and in the 200–2000 *m*/*z* mass range, using positive mode at the full scan spectra range 100 to 500 *m*/*z*, using the EC tune. Parameters were T 280 °C, ES 50 arb unit, Energy collision 29. The reference compound EC showed a characteristic peak at Rt interday 10.1 ± 1.5 and at Rt intraday = 9.4 ± 0.3 by LC-MS. The same chromatographic conditions were used for the LC-MS screening of NS, in order to compare it with EC-NS1 complex spectral data where an EC peak was detected (Appendix A, Appendix A).

### 3.9. Loading Efficiency and Loading Capacity of NS

The loading efficiency of NS was determined by an LC-SRM-MS (Selected Reaction Monitoring) study, which allowed us to quantify the amount of compound recovered in the EC-NS complex by monitoring the EC parent ion [M + H + Na]^+^, 315.1 *m*/*z*, and its daughter ion, 254.8 *m*/*z*. A calibration curve was obtained by using five solutions containing scalar concentrations of the reference pure EC (5 to 100 ppm, MeOH), which were analyzed in triplicate at 215 nm using the same chromatographic conditions described above for the reference EC. Experiments were performed on both the EC-NS1 and EC-NS2 extracts, the recovered EC (1.3 mg) of which was used to calculate both the loading efficiency and the loading capacity according to the following Equations (1) and (2), respectively [19]:Loading Efficiency = (μg Effective Drug Content/μg Theoretical Drug Content) × 100(1)
Loading Capacity = (μg Effective Drug Content/μg Loaded Complex) × 100(2)

## 4. Conclusions

NSs obtained by cross-linking β-CD residues with suitable reagents are nowadays acknowledged as valuable carriers for bioactive compounds [20]. In fact, many authors have demonstrated their usability to improve both the pharmacokinetic and pharmacodynamic profile of natural derivatives, including flavonoids and curcumin [21,22,23,24], as well as synthetic compounds such as anti-inflammatory, anti-tumor and antiviral drugs [25,26,27,28]. Compared to a single, even functionalized, β-CD unit, NSs show an improved ability to complex the hosted compound. Indeed, besides being included inside the hydrophobic cavity of the macrocyclic saccharide, the organic derivative may establish further interactions with the tridimensional net of the sponge. On the whole, the stability of the host-guest complex turns out to be increased. Moreover, the hosted compound is protected against the hydrolytic action of the surrounding environment.

In the present study, we proved that NSs obtained by cross-linking β-CD residues with diphenylcarbonate represent an excellent carrier also for the natural sesquiterpene EC, recovered from *E. crassus* SSt22 strain culture by an efficient lab-scale extraction. Moreover, spectroscopic and mass-spectrometric investigations showed a high grade of inclusion of the active compound within the NS, quantified by a 90% loading efficiency of the carrier. Significantly, the loaded compound was recovered unaltered from the complex. Indeed, once extracted from EC-NS and investigated by ESI-MS and ESI-MS/MS studies, it provided an analytical profile fully matching that of pure EC. As an inclusion complex with β-CD, EC can finally be unbiasedly investigated for its therapeutic potential, and hopefully used as a novel drug candidate from the marine world.

## Figures and Tables

**Figure 1 marinedrugs-20-00682-f001:**
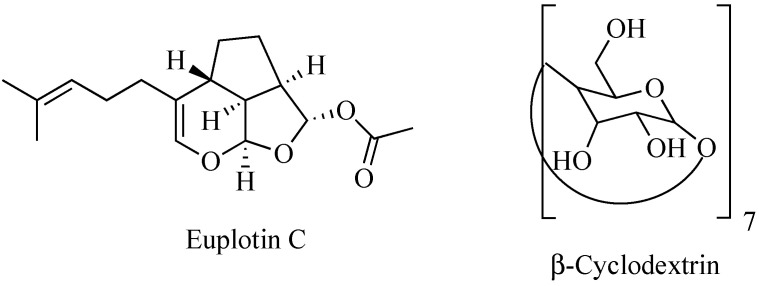
Chemical structures of euplotin C and β-cyclodextrin.

**Figure 2 marinedrugs-20-00682-f002:**
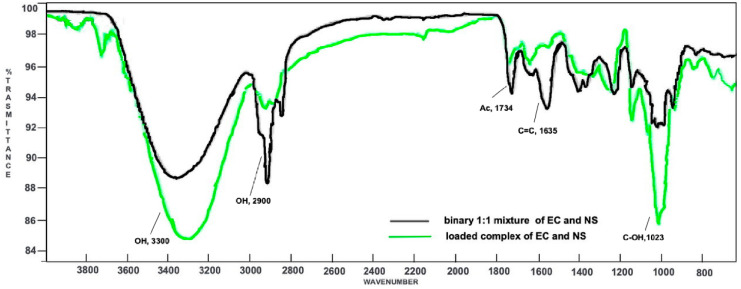
Superimposition of the ATR-FTIR spectra of the loaded complex EC-NS (green) and the binary 1:1 mixture of EC and NS (black).

**Figure 3 marinedrugs-20-00682-f003:**
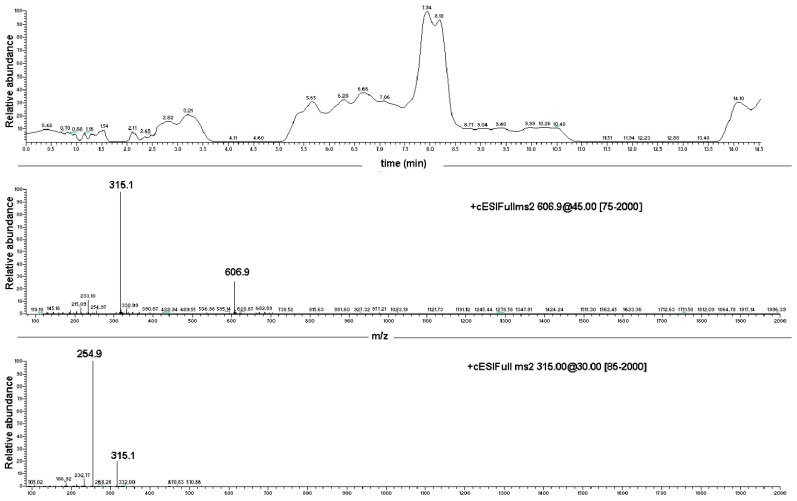
Direct infusion ESI-MS spectrum of EC, showing major ions at 606.9 *m*/*z* [2EC + Na^+^] and at 315.1 *m*/*z* [EC + Na^+^], and ESI-MS/MS spectrum of the EC-sodium adduct at 315.1 *m*/*z*.

**Figure 4 marinedrugs-20-00682-f004:**
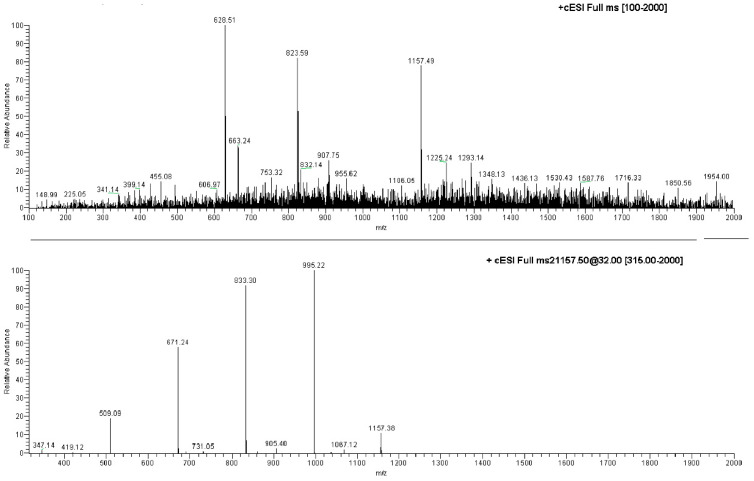
Direct infusion ESI-MS spectrum of NS and ESI-MS/MS spectrum on the ion at 1157.49 *m*/*z*.

**Figure 5 marinedrugs-20-00682-f005:**
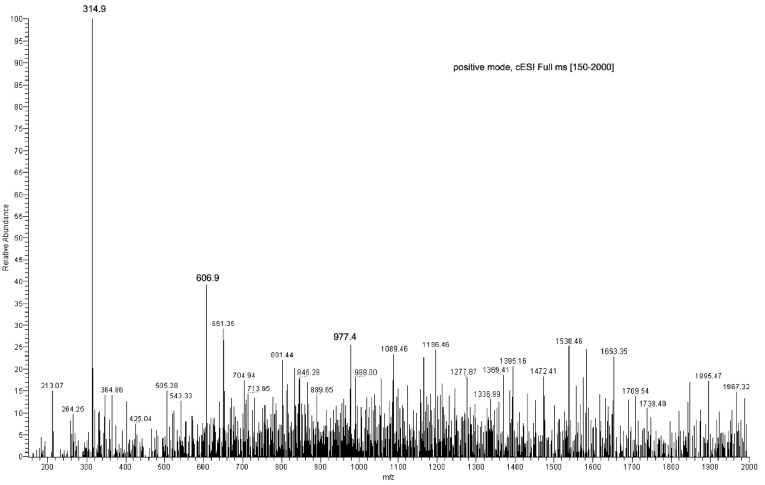
Direct infusion ESI-MS spectrum of EC-NS1 extract.

**Figure 6 marinedrugs-20-00682-f006:**
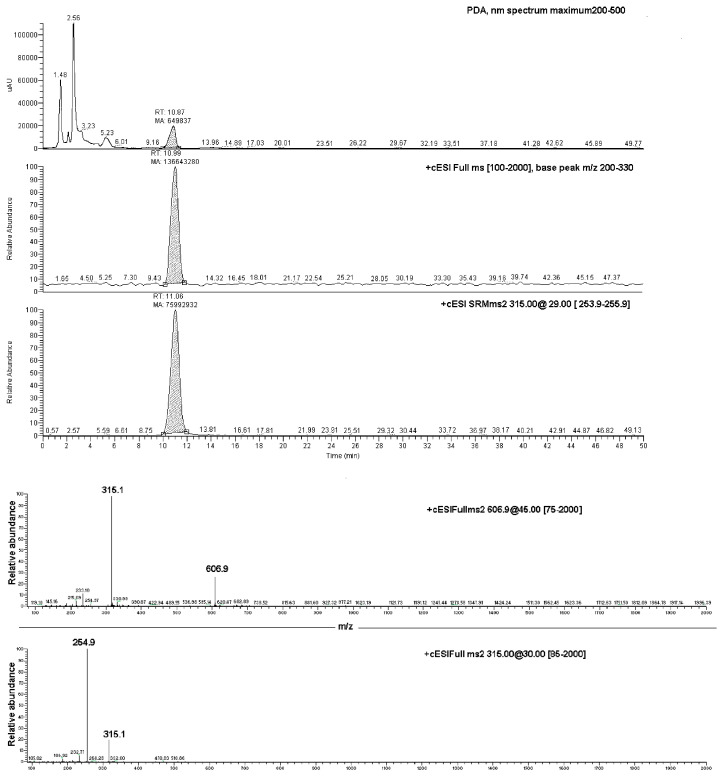
Upper box: LC-PDA-MS chromatograms of EC-NS1 extract and relative MS/MS peak detection at 606.9 *m*/*z* and 315.1 *m*/*z* for EC compound; lower box: ESI-MS/MS of the ion at 606.9 *m*/*z* (**top** panel) and ESI-MS/MS spectrum of the ion at 315.1 *m*/*z* (**bottom** panel).

## Data Availability

Not applicable.

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
