# Peer review of "Complexing the Marine Sesquiterpene Euplotin C by Means of Cyclodextrin-Based Nanosponges: A Preliminary Investigation"

_marinedrugs, 2022, doi:10.3390/md20110682_

Round 1

Reviewer 1 Report

The manuscript " Preserving the Chemical Stability of Euplotin C by Means of a Nano-structured Colloidal Vector: a Novel Opportunity to Vehiculate Marine Sesquiterpenoids" by Bertoli et al. deals with new β-cyclodextrin-based nanosponges of Euplotin C - a natural sesquiterpene class compound with high loading efficiency and stability. The newly obtained complex was prepared via crosslinking of β-cyclodextrin with diphenylcarbonate and was thoroughly characterized by various techniques.

Obviously, the obtained results are of great significance to the aspect of appropriate carriers for biologically active drugs. Investigating the classical solubilizing agent β-cyclodextrin (though quite preliminary) is also of interest. The manuscript is well written and structured, supported by necessary literature sources.  The study should be published in J Marine drugs. I have some recommendations and questions to the authors:

1. In the main manuscript ATR-FTIR spectra of the loaded complex EC-NS and the binary 1:1 mixture of EC and carrier are demonstrated. In the SI file it would be better to illustrate the ATR-FTIR spectra from Figs. S1, S2, S3, S4 in one figure under each other for better visualization.

2. The issue of poor solubility of  Euplotin C was underlined in the manuscript, and the opportunities of enhancing the solubility via β-cyclodextrin-based nanosponges was also mentioned. From this a question arises: why the solubility of the drug and β-cyclodextrin-based  samples was not studied? This information would be useful for pharmaceutics.

Reviewer 2 Report

The research presented in this manuscript is of great practical and scientific importance by describing an innovative method of obtaining and characterization of nanosponges based on β-cyclodextrin loaded with Euplotin C.

We recommend minor revisions when editing text, as follows:

- for the prefix nano-, all compound words are written connected (nanosponges, nanostructured, etc.);

- the subtitle of the article should be written with capital letter (...Colloidal Vector: A Novel Opportunity...) and no hyphenation;

- no period at the end for the title of the article;

- all e-mail addresses sould not be hyphenated;

- as the article is written in US English, the correct spelling is ‘tumor’; idem ‘colorless;

- line 28 – correct form – to stabilize;

- line 70 – correct form – we investigated the use;

- for the first entry of ‚nanosponges” also mark its acronym; therefore, for the entire text we will use only the acronym (NS); idem β-cyclodextrin (β-CD);

- line 98 – explanation for ATR-FTIR; idem for SPE, HPLC;

- line 143 – the correct reference entry is [7] or [8]? (by Guella and co-workers);

- lines 214-215 – correct form - solution (nanosponges and EC-loaded nanosponge);

- line 220 – use only acronyms;

- line 293 – correct form – eluting mixture;

- line 347 – correct form – daughter; idem within the supplementary material (×2).

Reviewer 3 Report

This is an interesting manuscript focused on the development of euplotin C and CD nanocomplexes as strategy to overcome the administration challenges of this drug. This manuscript is suitable for Marine Drugs Journal, as euplotin C is a sesquiterpene of marine origin. I consider that this is more than an interesting topic and I was excited to review it. It matches my research area. However, I have the feeling that the work is not complete. I mean, on technological part when developing DDS, more characterization is expected (size, PDI, Zeta potential, morphology, drug delivery). I think that the authors could also approach the manuscript from a biological point of view but in this case at least size and drug release profile are neccessary.  Efficacy studies  in melanoma could be also interesting. 

Major aspects:

1. Did other authors evaluate the use of nanocarriers for the administration of this drug?  If yes, it could be included in the introduction as it seems a little poor.

2. It could be interesting including information about nanosponges (hyperbranched cyclodextrin-based nanostructures) utility for drug delivery in the introduction.  In fact, lines 78-86 contains this information. This should be moved to introduction.

3. The authors should include the information of drug loading not just drug efficiency. 

4. Did the authors evaluate the particle size of nanosponges? Morphology by SEM or TEM? Drud release profile? Stability studies?

Minor points:

-          Abstract line 28: I suggest removing this sentence. Not provide much information as it is too general and common.

-          I suggest including data solubility in water of EC in the introduction.

-          Figure 2: Why did the authors not include the ATR spectra of single EC and CD?

Reviewer 4 Report

In the present manuscript, authors have synthesized b-cyclodextrin-based nanosponges and investigated their use as col-27 loidal carrier to stably complex euplotin C. The nanosponges were well characterized by various methods. The results also proved that the nanosponges can be loaded onto natural compound. The manuscript can be accepted after minor revision.

Provide the figures in better resolution.

Can authors comment on the particular choice of cyclodextrin?

Nanosponges size measurement studies should be reported.

Round 2

Reviewer 3 Report

I consider that the authors have improved a little the manuscript (introduction and some part of results). However, I am still think that the nanocomplex are poorly characterized. At least particle size and release should be done.  The proposed title " Preserving the Chemical Stability of Euplotin C by Means of a Nanostructured Colloidal Vector: A Novel Opportunity to Vehiculate Marine Sesquiterpenoids" confuse, as seems that you will find a full characterization of the  nanoformulation.  I suggest modifiying it.
